# The VNTR 48 bp Polymorphism in the *DRD4* Gene Is Associated with Higher Tobacco Smoking in Male Mexican Mestizo Smokers with and without COPD

**DOI:** 10.3390/diagnostics10010016

**Published:** 2019-12-30

**Authors:** Gloria Pérez-Rubio, Salvador García-Carmona, Leonor García-Gómez, Andrea Hernández-Pérez, Alejandra Ramírez-Venegas, Luis Alberto López-Flores, Raúl Sansores, Ramcés Falfán-Valencia

**Affiliations:** 1HLA Laboratory, Instituto Nacional de Enfermedades Respiratorias Ismael Cosio Villegas, Mexico City 14080, Mexico; glofos@yahoo.com.mx (G.P.-R.); chava_gc9@hotmail.com (S.G.-C.); llopezf92@gmail.com (L.A.L.-F.); 2Tobacco Smoking and COPD Research Department, Instituto Nacional de Enfermedades Respiratorias Ismael Cosio Villegas, Mexico City 14080, Mexico; leonor_garciag@hotmail.com (L.G.-G.); andrea.hde@gmail.com (A.H.-P.); aleravas@hotmail.com (A.R.-V.); 3Clínica de Enfermedades Respiratorias, Fundación Médica Sur, Mexico City 14080, Mexico; raulsansores@yahoo.com.mx

**Keywords:** VNTR, *DRD4*, tobacco smoking, genetic association

## Abstract

Cigarette smoking is influenced by nicotine’s effects on dopaminergic activity, which appear to be moderated by genetic variation, particularly a variable number tandem repeat (VNTR, 48 bp) polymorphism in the third exon of the dopamine receptor gene (*DRD4*). Smokers with the VNTR ≥7 repeats (long, L allele) report markedly increased participation in some smoking behaviors; hence, our aim was to evaluate the association of the L allele in Mexican Mestizo smokers with and without COPD. The *DRD4* VNTR 48 bp was genotyped in 492 Mexican Mestizo smokers: 164 COPD patients (≥20 cigarettes per day, cpd), 164 heavy smokers without COPD (HS, ≥20 cpd) and 164 light smokers without COPD (LS, 1–10 cpd). In the dominant model analysis (SL + LL vs. SS), men in the COPD and HS groups showed a statistical difference compared to LS (*p* = 0.01, OR = 2.06, CI 95% 1.17–3.64 and *p* = 0.05, OR = 1.88, CI 95% 1.03–3.45, respectively). In addition, by clustering smokers >20 cpd (COPD + HS) and comparing with the LS group, we found an association with increased risk of higher tobacco smoking *p* = 0.01, OR = 1.99, CI 95% 1.18–3.34. In conclusion, the long allele (L) in the VNTR of the *DRD4* gene is associated with the risk of presenting higher tobacco smoking in male Mexican Mestizo smokers.

## 1. Introduction

In Mexico, the 2016–2017 National Survey on Drug, Alcohol and Tobacco Use reported that 17.6% of the population aged 12 to 65 smoked tobacco; of these, 12.3% presented nicotine dependence; in 2017, only 16.3% of these smokers had abandoned this consumption [1]. The relationship between chronic obstructive pulmonary disease (COPD) and cigarette smoking as the major risk factor, in addition to genetic factors, has been extensively reviewed [2].

Cigarette smoking is influenced by nicotine’s effects on dopaminergic activity. The increase of dopamine in the nucleus accumbens is considered to be one of the first mechanisms that participate in the reward and maintains the reinforcement towards drug use [3]. Among the widely studied dopamine receptors is D4, encoded by the *DRD4* gene, which is located near the telomere on the short arm of Chromosome 11, and has four exons. In Exon 3, there is a polymorphism of variable number tandem repeats (VNTR) of 48 bp, which can range from 2 to 11 repeats [4]. Repeats of two, four, and seven are the most frequent in human populations. In general, to facilitate their study, individuals who have six or fewer repeats are known as carriers of the short allele (S), and those with seven or more repeats as carriers of the long allele (L) [5]. The presence of the L allele decreases the expression of the gene in vitro and, if dopamine is bound to this receptor, cyclic adenosine monophosphate (cAMP) formation decreases [6]. Individuals who smoke and carry at least one L allele exhibit significantly greater cravings compared to those with homozygous SS [7]; individuals with the L allele show greater brain activity in the right insula and in the upper frontal gyrus when they observe images related to cigarettes. Similarly, the L allele has been associated with a low prevalence rate in smoking cessation [8]. No studies looking for the participation of VNTR 48 bp in the *DRD4* gene in the COPD susceptibility have been conducted to date. The aim of this study was to evaluate the association of the L allele with smoking in smokers with and without COPD.

## 2. Materials and Methods

### 2.1. Study Population

A prospective and exploratory study was conducted, for which 492 participants were recruited as follows: 164 patients with COPD secondary to tobacco smoking, smoking at least 20 cigarettes per day (cpd); 164 heavy smokers (HS, ≥20 cpd); and 164 light smokers (LS, 1–10 cpd). The latter two groups included smokers with no evidence of pulmonary disease and with normal spirometry parameters. All participants were invited to participate voluntarily and signed an informed consent letter. The protocol was previously approved by the Institutional Committee of Biosafety, Research, and Research Ethics (approval code B15-16) of the Instituto Nacional de Enfermedades Respiratorias Ismael Cosío Villegas (INER). Included subjects attended COPD and smoking cessation support clinics, both part of the Department of Smoking and COPD Research Department of the INER, Mexico.

All participants underwent a background questionnaire related to inherited pathologies, whereby subjects who reported suffering some type of lung (other than COPD) and/or chronic inflammatory disease were excluded, as were subjects with non-Mexican ancestry (with no Mexican-birth parents or grandparents).

The COPD diagnosis was confirmed using lung function tests, considering a ratio of forced expiratory volume in the first second/forced vital capacity (FEV_1_/FVC) <70% post-bronchodilator as COPD, according to the reference values for Mexicans [9]. Subjects diagnosed with bronchial asthma, bronchiectasis, active tuberculosis, lung cancer, cystic fibrosis, hypersensitivity pneumonitis, or idiopathic pulmonary fibrosis were excluded.

### 2.2. Sample Size

The sample size was calculated considering the frequency of 7 repeats (25.5%) in a Mexican population [10], considered using a two-tailed level of confidence of 95%, with a statistical power of 80% and a ratio of one case for each control. Under these conditions and using the Epi Info 7.1.5.2 program [11], the result obtained was 164 subjects for each comparison group.

### 2.3. DNA Extraction

The DNA was extracted from peripheral blood cells via venipuncture, using the commercial BDtract Genomic DNA isolation kit (Maxim Biotech, San Francisco, CA, USA). The DNA was then quantified by UV absorption spectrophotometry at the 260 nm wavelength using a NanoDrop system (Thermo Scientific, Wilmington, DE, USA).

### 2.4. Genotyping

Specific primers were designed for the VNTR located in Exon 3 of the *DRD4* gene; for this, the Primer-BLAST tool was used [12]. The primers obtained were Forward: GCG ACT ACG TGG TCT ACT CG and Reverse: GTG CAC CAC GAA GAA GGG.

End-point polymerase chain reaction (PCR) amplification was carried out with 25 µL of a reaction consisting of 100 ng/µL of genomic DNA, 1X buffer (cat. KK1510, Kappa Biosystems, Wilmington, MA, USA), 0.5 µM of each primer (Applied Biosystems, Foster City, CA, USA), 1 mM MgCl_2_ (cat. KK1513, Kappa Biosystems, MA, USA), 2.5 IU KAPA Taq HotStart (cat. KK1507, Kappa Biosystems, MA, USA), 5% dimethyl sulfoxide (DMSO) (cat. W387520, Sigma, St. Louis, MO, USA), 200 µM dATP, dCTP and dTTP (cat. 10297-018, Invitrogen, Waltham, MA, USA), 140 µM dGTP (cat. 10297-018, Invitrogen, MA, USA) and 60 µM 7-deaza-2′-deoxy-guanosine-5′-triphosphate (cat. 10988537001, Sigma, MO, USA).

The following cycling conditions were used to carry out the PCR reaction: 1 cycle: 3 min, 95 °C; 35 cycles: 30 s 95 °C, 30 s 55 °C, and 1 min 72 °C; Finally, a 1 min cycle at 72 °C. The above was carried out using a Veriti 96-Well Thermal Cycler instrument (ThermoFisher Scientific, Waltham, MA, USA).

The amplification PCR products were then electrophoresed in submarine 2% agarose gels containing 0.2 µg/mL ethidium bromide (cat. E1385, Sigma-Aldrich, Darmstadt, Germany) for 40 min (30 V/cm) and amplified bands were visualized in a dual intensity UV light transilluminator (UVP Inc., Upland, CA, USA) before being stored and evaluated in the Electrophoresis Documentation and Analysis System (EDAS 290) (Eastman Kodak, New Haven, CT, USA).

A molecular weight marker of 100 bp (cat. CSL-MDNA-100BPH, Cleaver Scientific United Kingdom) was used; amplification products with weight ≤ 740 bp were classified as carriers of the short allele (SS), while those with product of 780 bp or higher were classified as carriers of the long allele (LL), and those samples with two bands between the previously described weights were characterized as heterozygous (SL).

### 2.5. Statistical Analysis

Before performing the genotypic comparisons, the Hardy–Weinberg equilibrium (HWE) of polymorphism was calculated. For each quantitative variable, the Kolmogorov–Smirnov test was performed to assess normality; parametric statistics were evaluated for those variables with normal distribution, while for those with free distribution, non-parametric statistics were evaluated. This analysis was performed using SPSS version 15 (SPSS software, IBM, New York, NY, USA). For the qualitative variables, Pearson’s chi-square or Fisher’s exact test was performed as appropriate in a contingency table. For the quantitative variables, a Pearson’s or Spearman’s correlation was made according to their distribution [13]—this correlation and the representation of the results were carried out using the ggplot2 package [14] was used in Rstudio [15]. The analysis of alleles and genotypes was performed using a chi-square (χ^2^) test with Yates correction, using the EPIDAT program version 3.1 [16]. A logistic regression analysis was carried out to adjust by potential confounding variables using Plink v. 1.07 [17].

## 3. Results

### 3.1. Clinical and Demographical Variables

Table 1 shows the demographic variables of the subjects included in the present study. Each of them had a free distribution; the group of patients with COPD had more years of smoking (43 years) compared to the groups of heavy and light smokers (36 and 29 years respectively). Lung function (evaluated by FVC, FEV_1_, and FEV_1_/FVC parameters) was lower in the COPD group compared to the other two study groups; however, performing the Mann–Whitney U test between the HS and LS groups returned no statistically significant difference (*p* = 0.062). The age and weight between the HS and LS groups did not show statistical differences (*p* = 0.158 and *p* = 0.233, respectively). The same was found for height between the COPD and HS groups (*p* = 0.144), cpd (*p* = 0.082) and age of smoking onset (*p* = 0.637). COPD patients had a lower body mass index among groups (*p* = 0.001). In the Fagerström test, the COPD and HS groups showed no statistically significant difference (*p* = 0.706).

The Spearman test was used to analyze the correlation between quantitative variables of the three groups included in the study. Figure 1 proves the correlogram between the variables studied for the whole study population, which showed a low negative correlation (−0.30 to −0.50) between years of smoking and the FEV_1_, FVC, and FEV_1_/FVC post-bronchodilator values, and the same tendency with cpd.

### 3.2. Allele Association and Genotype Analysis

The genotype frequencies in the study populations met with the Hardy–Weinberg equilibrium (*p* = 0.56), which is a principle stating that the genetic variation in a population will remain constant from one generation to the next in the absence of disturbing factors [18]. Table 2 shows the allele and genotype frequencies of the study groups. The L allele is more frequent in the COPD and HS groups (35.06% and 32.31% respectively, *p* = 0.508), however, these differences are not statistically significant (*p* = 0.244) when compared to the LS group (30.48%). As for the homozygous LL genotype, it is found in a higher frequency in the COPD group (13.40%) but, there are no significant differences when compared with the HS and LS groups (10.36% and 12.19%, *p* = 0.463 and *p* = 0.233 respectively).

Allele frequencies reported in the present study were similar to reports made in a clinically healthy population of Mexico City and those reported in Hispanics. (Figure 2).

We did not find any association with alleles and genotypes, nor with dominant nor recessive inheritance models (*p* > 0.05, Appendix A).

In the logistic regression of the HS vs. LS comparison, we used height, years smoking, cpd, age of onset, and Fagerström score as covariates. For the analysis between COPD vs. HS, we used age, weight, and years smoking; while for COPD vs. LS comparison, in addition to the covariates mentioned in the previous comparison, we used height, cpd, age of onset, and Fagerström score. The results obtained from these analyses were not statistically significant.

Later, clustering subjects from all groups, we made comparisons stratifying by smoking age of onset (<18 vs. ≥18 years old). Codominant, dominant, and recessive models were applied and then divided according to sex. No statistically significant differences were found in these comparisons (Appendix A).

Finally, comparisons were carried out by dividing each study group according to sex. Interestingly, in the dominant model analysis, women reached almost 59% of the genotypes harboring one or two copies of L allele (SL + LL) among light smokers in comparison to men (~41%), making this difference statistically significant (*p* < 0.05). No other differences were found in these comparisons (Appendix A), or when evaluating a codominant and dominant model among study groups (Appendix A).

However, among men, the COPD vs. LS comparison was found to be significantly associated with risk (dominant model, *p* = 0.01, OR = 2.06, CI 95% 1.17–3.64) when the L allele was heterozygous or homozygous (Table 3).

Under this same model, comparing HS vs. LS showed a risk trend (*p* = 0.055, OR = 1.88, 95% CI 1.03–3.45). Clustering smokers > 20 cpd (COPD + HS) and comparing with LS group, we found an association with increased risk of higher tobacco smoking (*p* = 0.01, OR = 1.99, CI 95% 1.18–3.34). The codominant and dominant models comparing COPD vs. HS did not show significant differences (Appendix A).

## 4. Discussion

Several genetic variants have been reported to be associated with nicotine addiction, suggesting the existence of a genetic component; however, few studies in populations other than the Caucasian have explored genetic variants in the dopaminergic pathways. In the *DRD4* gene, most studies have focused on single-nucleotide polymorphisms (SNP) and their association with nicotine addiction; previous studies have suggested that alleles of the *dopamine receptor D4* gene moderates the effect of nicotine on cigarette reward [19], greater craving [20], and neuroticism [21] among smokers; however, there have been no studies evaluating susceptibility to tobacco smoking/nicotine addiction to date.

The aim of the current study was to investigate genetic susceptibility to an increase in the cigarettes per day smoked in the presence of the VNTR L allele (≥7 repeats) in the *DRD4* gene in Mexican Mestizo smokers with and without COPD. The COPD group was the oldest and presented most years of smoking continuously; this was directly reflected in the score obtained in the Fagerström test, since it was the group with the highest degree of nicotine addiction and worst lung function. The HS and LS groups showed significant differences only in cpd, age of onset, and punctuation in Fagerström. Additionally, we identified a negative and significant correlation between higher number of years smoking and decreased values of FEV_1_, FVC, and FEV_1_/FVC; this same pattern was identified when evaluating cpd.

The results of the genetic analysis demonstrated no association between the presence of the 7 repeat allele in the *DRD4* gene and higher cigarette consumption, or among those patients with a diagnosis of COPD secondary to smoking where nicotine addiction (evaluated by Fagerström test) was higher. The frequencies of the short (S) and long (L) alleles in the study population were similar to the rest of the populations previously reported in the literature [10,22,23,24,25].

The *DRD4* gene codes for a dopamine receptor consisting of 387 amino acids with seven transmembrane domains [26]. It has four exons, and the VNTR polymorphism of 48 base pairs is located in the third of them, which encodes the third intracellular domain, which can have from 32 to 176 amino acids depending on the number of repeats present in the gene [4]. The 7 repeat allele has been associated with a low response to drug-stimulated dopamine [27] and lower neurotransmitter release after tobacco smoking [28].

In our study, we identified that the men of the COPD vs. LS comparison presented an increased risk association (OR > 2) when the L allele was present in a heterozygous or homozygous state. In the comparison of HS vs. LS, the same trend was observed (*p* = 0.05). We hypothesize that this association is due to the nicotine addiction more than COPD itself, since variables related to tobacco smoking (cpd, age of onset) were very similar in the COPD and HS groups.

It is well known that women and men begin smoking for different reasons and that quitting smoking is influenced by different factors (stress, anxiety, and depression, among other factors). At the cerebral level, through images obtained by positron emission tomography, differences have been observed, mainly in the ventral striatum. Between sexes, there are also spatial and temporal differences in dopaminergic signaling [29]. There have been studies where differences have been described according to sex in the ventral striatum and in the availability of dopamine D2/D3 receptors in smokers [30]. However, in a Caucasian population subject to nicotine replacement therapy, at 26 weeks of monitoring, no statistically significant differences were observed when comparing SS vs. SL + LL individuals [31]. Later, in a smoker population of Caucasian origin, carriers of the L allele were kept in abstinence (monitored for up to one year) when treated with bupropion compared to those with the S allele [32]. That same year, another study was published examining Caucasians of Eastern Europe, where they found no association between the 7 repeat allele and being a smoker [33]. The results on the presence of 7 repeat allele and the association with nicotine addition, cigarettes per day, and other phenotypes related to smoking are contradictory. This is evidence of the complexity of nicotine addiction and the involvement of the dopaminergic pathway at the genetic level.

There are more than 200 polymorphisms in the *DRD4* gene; according to their biological relevance and location in the gene, SNPs located in the promoter region (rs936462, rs3758653, and rs1800955) [34] have been studied. In our workgroup, only rs1800955 was found to be associated with the risk of smoking in carriers of the CC genotype [35].

To our knowledge, this is the first published study of genetic participation of the *DRD4* VNTR 48 bp on tobacco smoking; previous studies have focused on particular phenotypes and/or cessation variables [7,20,36], mostly in Caucasian populations.

The current study is particularly important due to the miscegenation (from Caucasian and Amerindian) present in Latin-American populations such as those from Mexico. Comparative analyses between Mexicans and other neighboring populations have revealed significant differences in genetic diversity [37]. It is important to evaluate the role of the *DRD4* gene in the mechanism of nicotine addiction; however, it is also necessary to consider, in addition to SNP and VNTR polymorphisms, other mechanisms which could participate in the risk of nicotine addiction, such as epigenetic factors.

This study is not exempt from limitations. For example, the sample size was reduced in comparison to other studies, and our participants were recruited from a single center. Studies including other centers would be desirable. Although the Fagerström test is a suitable instrument to evaluate nicotine addiction, cigarette consumption has a very particular context; each consumer has their style for smoking (puff number/volume, puff interval, and duration), and we were not able to analyze blood or urine cotinine levels to ascertain the self-reported cpd smoked by the subjects, to know exactly how much nicotine entered the body and have a more objective measurement of the degree of addiction.

## 5. Conclusions

The long allele (L) in the VNTR of the *DRD4* gene is associated with the risk of presenting higher tobacco smoking in male Mexican Mestizo smokers.

## Figures and Tables

**Figure 1 diagnostics-10-00016-f001:**
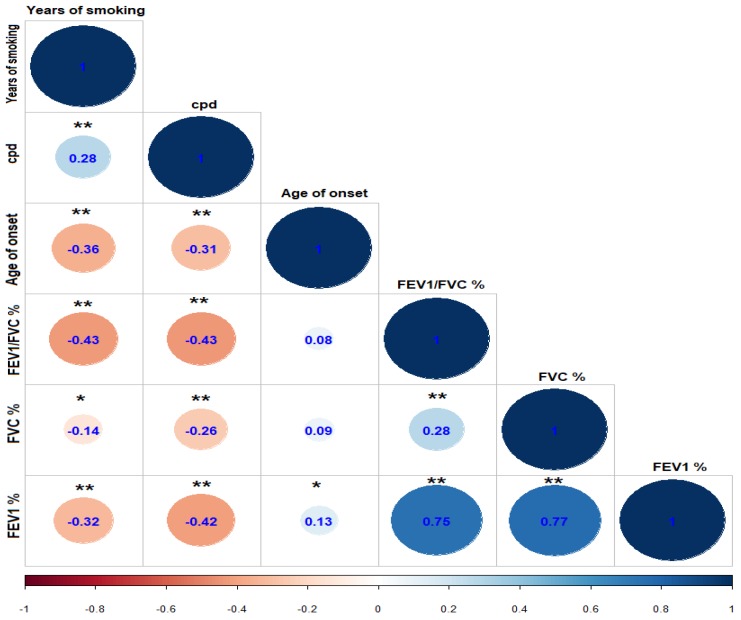
Correlogram between quantitative variables of the three groups included in the study, showing Spearman’s rho. * *p* < 0.05. ** *p* ≤ 0.01. FEV_1_, FVC, and FEV_1_/FVC values are shown post-bronchodilator. Abbreviations: cpd: cigarettes per day; FVC: forced vital capacity; FEV_1_: forced expiratory volume in the first second.

**Figure 2 diagnostics-10-00016-f002:**
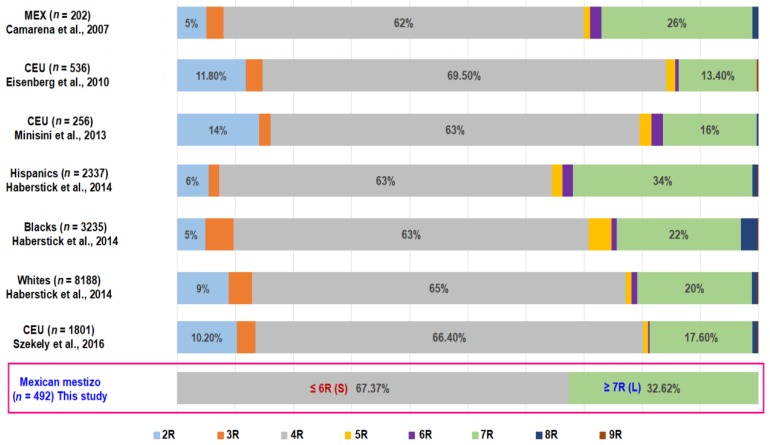
Allele frequencies of VNTR in *DRD4* in seven human populations and in the current study.

**Table 1 diagnostics-10-00016-t001:** Demographic variables of the subjects included in the present study.

Variable	COPD *n* = 164	HS *n* = 164	LS *n* = 164	*p*-Value *
Age (years)	59 (45–73)	53 (30–75)	50 (35–75)	<0.001
Weight (kg)	69 (46–91)	70 (45–118)	70 (50–109)	0.003
Height (m)	1.63 (1.40–1.88)	1.65 (1.43–1.86)	1.60 (1.40–1.88)	0.002
BMI	25.7 (16.3–41.3)	26.9 (15.9–46.6)	27.1 (17.2–53.6)	0.001
Years of smoking	43 (28–60)	36 (5–62)	29 (3–52)	<0.001
cpd	20 (20–80)	20 (20–40)	8 (1–10)	<0.001
TI	43 (28–239)	36 (5–123)	12 (0.1–26)	<0.001
Age of smoking onset	17 (7–30)	15 (9–35)	18 (12–45)	<0.001
Fagerström test	7 (0–9)	6 (1–10)	3 (0–8)	<0.001
FVC (%)	82 (55–135)	93 (54–133)	100 (71–128)	<0.001
FEV_1_ (%)	68 (31–112)	92 (50–128)	101 (67–131)	<0.001
FEV_1_/FVC (%)	60 (44–81)	80 (58–101)	83.7 (64–97)	<0.001

* *p*-value by Kruskal–Wallis test. FEV_1_, FVC, and FEV_1_/FVC values are shown post-bronchodilator. Abbreviations: cpd, cigarettes per day; BMI, body mass index; TI, tobacco index; FVC, forced vital capacity; FEV_1_, forced expiratory volume in the first second; COPD: chronic obstructive pulmonary disease; HS, heavy smokers; LS, light smokers.

**Table 2 diagnostics-10-00016-t002:** Allele and genotype frequencies of VNTR in *DRD4* among study groups.

*DRD4*	COPD (*n* = 164)		HS (*n* = 164)		LS (*n* = 164)		*p*-Value		
	*n*	AF/GF%	*n*	AF/GF%	*n*	AF/GF%	COPD vs. HS	COPD vs. LS	HS vs. LS
**S**	213	64.93	222	67.68	228	69.51	0.508	0.244	0.674
**L**	115	35.06	106	32.31	100	30.48			
**SS**	71	43.30	75	45.73	84	51.21			
**SL**	71	43.30	72	43.90	60	36.58	0.463	0.233	0.625
**LL**	22	13.40	17	10.36	20	12.19			

Abbreviations: COPD: chronic obstructive pulmonary disease; HS: heavy smokers; LS: light smokers; AF: allele frequencies; GF: genotype frequencies; S: short allele (*DRD4* VNTR 48 bp ≤ 6 repeats); L: long allele (DRD4 VNTR 48 bp ≥ 7 repeats); SS: homozygous short allele; SL: heterozygous (one short allele and one long allele); LL: homozygous long allele.

**Table 3 diagnostics-10-00016-t003:** The dominant model in men included in this study.

Men	COPD (*n* = 130)		LS (*n* = 80)		*p*-Value	OR (CI 95%)
	n	F%	n	F%		
SS	53	40.7	47	58.8	0.0168	
SL + LL	77	59.2	33	41.2		2.06 (1.17–3.64)
	**HS (*n* = 93)**		**LS (*n* = 80)**			
	*n*	F%	*n*	F%		
SS	40	43.0	47	58.8	0.0559	
SL + LL	53	57.0	33	41.2		1.88 (1.03–3.45)
	**COPD + HS (*n* = 223)**		**LS (*n* = 80)**			
	*n*	F%	*n*	F%		
SS	93	41.7	47	58.8	0.0127	
SL + LL	130	58.3	33	41.2		1.99 (1.18–3.34)

Abbreviations: COPD: chronic obstructive pulmonary disease; LS: light smokers; F: frequency; CI: confidence interval at 95%; S: short allele (*DRD4* VNTR 48 bp ≤ 6 repeats); L: long allele (*DRD4* VNTR 48 bp ≥ 7repeats); SS: homozygous short allele; SL: heterozygous (one short allele and one long allele); LL: homozygous long allele. *p*-value by Yate’s correction.

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
