# Peer review of "The VNTR 48 bp Polymorphism in the DRD4 Gene Is Associated with Higher Tobacco Smoking in Male Mexican Mestizo Smokers with and without COPD"

_diagnostics, 2019, doi:10.3390/diagnostics10010016_

Round 1

Reviewer 1 Report

Congratulations on the job The tables could be improved. Explain the abbreviations used in the tables

Author Response

Thank you very much for your kind comment. Now tables contain the properly abbreviations requested in each one.

Reviewer 2 Report

Page 1: Please explain the term "Mexican mestizo" smokers, for international readers". Are they different from other mestizo? Page 2: line 45. Figure 1. There is no figure 1 in the text, but there two figure 2. Please correct it. Table 1: Can you add some data about Body Mass Index? You have age and weight. Page 2: line 61-64. Information not found in the Table. Page 2: line 60-61: lung function? There are data not in the table Page 2: intensity of smoking is defined by pack x years. Can you add this information in the text? Page 2: line 74. It should be reformulated for clear understanding. Number are different. Page 3, figure 2. Unusual template, with a lot of information. Good, but maybe you can add some explanation? Page 3: line 83: you should explain the Hardy-Weinberg equilibrium. Page 5: line 135: Not clear, can you reformulate? Page 6: Line 192. 4. Materials and Methods. Not before results? Page 7: line 217. Bibliography number 30 not in the text? Page 10: line 360. 30 not in the text?

Author Response

Author response: Thank you for the opportunity to improve our manuscript.

Page 1: Please explain the term "Mexican mestizo" smokers, for international readers". Are they different from other mestizo?

Author response: The present-day Mexican population is the result of a process of miscegenation between native Amerindian, Spanish and African populations; the resulting genetic recombination of these populations allowed the emergence of new mestizo populations. Attending your question, now we have integrated a small paragraph in the discussion section describing the existence of differences in mestizos from Mexico and other Latin-American populations, including the appropriate reference.

Page 2: line 45. Figure 1. There is no figure 1 in the text, but there two figure 2. Please correct it.

Author response: The first "figure two" has been renumbered as figure 1, and the mention in the line 45 has been deleted.

Table 1: Can you add some data about Body Mass Index? You have age and weight.

Author response: Body mass index was added in table 1 and described in the text.

Page 2: line 61-64. Information not found in the Table. Page 2: line 60-61: lung function? There are data not in the table

Author response: Regarding lung function concerns, this has been solved including a brief line indicating that these data refer to FVC, FEV1, and FEV1/FVC parameters.

Page 2: intensity of smoking is defined by pack x years. Can you add this information in the text?

Author response: Tobacco index value, including packs/years, was added in Table 1.

Page 2: line 74. It should be reformulated for clear understanding. Number are different.

Author response: Text in line 74 has been reworded, to better understanding.

Page 3, figure 2. Unusual template, with a lot of information. Good, but maybe you can add some explanation?

Author response: Correlogram has been described precisely in the previous paragraph to the corresponding figure.

Page 3: line 83: you should explain the Hardy-Weinberg equilibrium.

Author response: Hardy-Weinberg equilibrium principle was explained, and the corresponding reference is included.

Page 5: line 135: Not clear, can you reformulate?

Author response: Done.

Page 6: Line 192. 4. Materials and Methods. Not before results?

Author response: Thank you for the advice; now has been modified, tables and figures numbers, as well as references, have been reordered.

Page 7: line 217. Bibliography number 30 not in the text?

Author response: Yes, this is in the methods section, 4.4 Genotyping subsection. Actually, according to the new order, all references have been reordered.

Page 10: line 360. 30 not in the text?

Author response: Now has a different number, but yes, it is included. Now is the reference #12.